# Allegro: GPU Simulation Acceleration for Machine Learning Workloads

Euijun Chung, Seonjin Na, Hyesoon Kim

Georgia Institute of Technology

*euijun@gatech.edu, seonjin.na@gatech.edu, hyesoon@cc.gatech.edu*

*Abstract*—Current GPU simulators face challenges handling large machine learning workloads like LLMs due to their slow execution. To address this issue, we leverage our observations on the massive number of GPU kernel calls inside these workloads. Given the homogeneous nature and cache-unfriendly behavior of these kernels, we demonstrate that they exhibit identically and independently distributed (i.i.d.) execution times, thus allowing us to apply statistical sampling approaches for accurate GPU kernel sampling. In this paper, we introduce Allegro, a practical methodology for GPU simulators aimed at substantially reducing execution time while maintaining high accuracy comparable to actual execution. Employing a statistical measure with a recursive algorithm, we design an accurate kernel sampling scheme, supported by a proof of theoretical error bounds. By integrating Allegro into Macsim, we achieve a simulation speedup of 983.96x on 7 of the latest ML workloads with an error rate of 0.057%. Compared to the random sampling method, this average error is 9.22x smaller with the speedup value fixed. Additionally, we demonstrate that adjusting the error bound enables the simulator to achieve greater speed enhancements with only a marginal rise in error rate. This adaptability offered by Allegro allows researchers to easily balance desired performance and accuracy.

## I. INTRODUCTION

Given the widespread adoption of large-scale Machine Learning (ML) models and their applications, such as Large Language Models (LLMs), their significant computational demands have become a crucial problem [29]. Many researchers are actively seeking methodologies to optimize the hardware architecture of accelerators tailored for machine learning workloads [21], or to resolve the hidden performance bottlenecks in software perspective [8]. GPU's extensive thread-level parallelism has made them become the predominant accelerator for the ML workloads [24]. Consequently, researchers are actively exploring faster and more energy-efficient GPU architectures to meet the evolving demands of ML tasks.

The need for a precise and fast architectural simulator for GPUs is also rising along with their demands. Architectural simulators play an important role in both validating new architectural designs and finding the hidden performance bottlenecks in the processing pipeline. Macsim [23], AccelSim [22], and MGPUSim [37] are the most widely used GPU simulators. However, without any optimization techniques applied to the simulators, they suffer from significant slowdowns on running ML workloads, especially on the workloads using the latest Deep Neural Network (DNN) models [2]. Table I shows the relative slowdowns on these simulators compared to the real GPU. Running the latest ML workloads on a simulator is

TABLE I
THROUGHPUT AND SLOWDOWN OF GPU SIMULATORS.

|  | Real GPU | Macsim | GPGPU-Sim | MGPUSim |
|---|---|---|---|---|
| Simulation Rate (KIPS) | 4103750 | 50.5 | 12.5 | 27 |
| Relative Throughput | 328300 | 4.04 | 1 | 2.16 |
| GPT-2: Generate 100 tokens | 0.925 sec | **20.88 hrs** | **3.52 days** | **1.63 days** |

an important issue, as reducing the discrepancy between the simulated and actual GPU usage is critical for accurately validating new architecture designs.

A few previous works have been made to speed up architectural simulations. However, methodologies designed for CPU simulations [5], [13], [33] do not apply to GPU simulations due to the incomparable amount of thread parallelism inherent in GPUs. Studies on GPU simulations [2], [25], [28] have indeed achieved speedups in GPU workloads. However, they do not provide theoretical bounds for sampling error nor they do not leverage the characteristics of GPU kernels utilized in ML workloads to achieve a greater degree of speedup.

In this paper, we begin by analyzing the newest ML workloads, specifically focusing on state-of-the-art large language models (LLMs), to identify key insights that can be leveraged to enhance simulation acceleration. Subsequently, we introduce **Allegro**, a statistic-based sampling approach designed to accurately reduce the GPU simulation time of ML workloads. In summary, our work makes the following contributions:

- This paper presents novel insights into the characteristics of the latest ML workloads on GPUs, highlighting that GPU kernels for ML exhibit high homogeneity and identically and independently distributed (i.i.d.) execution times, enabling the adoption of statistical approaches such as the Central Limit Theorem (CLT) and other sampling methods.
- We present methodologies demonstrating how statistical theories can be applied to design a simple and accurate measure, which we then utilize to propose Allegro's sampling algorithm. Additionally, we provide mathematical proofs concerning the bounds of error in our approach.
- By executing 7 latest ML workloads, we validate that Allegro achieves speedup with errors falling within the given error bound and smaller than previous works. The

achieved speedup reaches a maximum of 1486.66x, with an average speedup of 983.96x, while maintaining the error within 0.057%. Moreover, we integrated Allegro into Macsim and successfully completed ML workloads.

## II. BACKGROUNDS

### A. Limitations of Current GPU Simulators

Table I presents the simulation rate and the relative slow-down of various GPU simulators, along with the results from a real GPU. The setup environment for experiments is shown in Section V-A. The GPU simulator statistics are sourced from papers (GPGPU-Sim and MGPUSim) [22], [37] or measured directly (Macsim). Even though generating a single 100-token length sentence with a GPT-2 model takes less than 1 second on a real GPU, the same workload on a GPU simulator takes around a few days, as shown in Table I. It is infeasible to run GPU simulators with the latest ML workloads due to these long simulation times. Therefore, performance optimization techniques for GPU simulators are necessary.

### B. Works on Accelerating Architectural Simulations

There have been numerous efforts to address this performance slowdown. Works such as Simpoint [13], Barrier-point [5], and similar methods [33] propose techniques to accelerate CPU simulations. However, existing CPU solutions are not applicable in the GPU domain due to the massive level of thread parallelism of GPUs [2].

Similar approaches optimized for GPUs have also been proposed, including TBPoint [19], PKA [2], Photon [25], and Sieve [28]. We categorized these solutions as well as other GPU performance modeling techniques as follows.

**Analytical models** ([15], [16], [18], [27], [42]): Building and employing an analytical model for evaluating GPU architectures offers a fast and efficient approach. However, such models may lack accuracy compared to cycle-level simulations, as ensuring the accuracy of analytical models requires regular updates to follow up on the design changes in new GPU architectures. Additionally, such analytical models are not capable of reflecting the impact of certain architectural design changes, such as different cache replacement policies or workload scheduling policies (e.g., warp and block scheduling in GPUs).

**Early stopping** ([2], [28]): Early stopping simulations after stabilization of IPC is another approach that can yield speedup in simulations. However, in many GPU workloads, it is known that IPCs do not remain stable during the entire workload execution [25], and this may lead to significant errors in the simulation results.

**Workload Sampling** ([2], [19], [25], [28]): Workload sampling in GPU simulation has been a popular solution due to its effectiveness and acceptable accuracy. TBPoint [19] is the first paper to utilize clustering and sampling for GPU kernels to achieve efficient and accurate sampling. However, this method requires per-application tuning and does not provide sufficient speedup for running modern ML workloads.

PKA [2] is a follow-up work that also employs cluster-then-sample and early-stopping techniques on GPU kernels. Although this work demonstrates that GPU kernel sampling can yield simulation speedup for various GPU workloads, it lacks sufficient speedup to run ML workloads and lacks any theoretical explanations of the sampling errors.

Photon [25] and Sieve [28] are subsequent works after PKA [2], and they demonstrate notable speedup on certain workloads. However, these works still rely on empirical kernel classification and sampling techniques.

To address these issues, we propose Allegro, an efficient and accurate kernel sampling methodology for ML workloads. We first focus on ML workloads and present several observations indicating that statistical approaches can be applied to model the sampling process. In Section III, we demonstrate that the GPU kernels used in ML workloads are homogeneous and have identically and independently distributed (i.i.d.) execution times. Subsequently, we design an algorithm and a sampling scheme that can accurately and effectively sample the GPU kernels by leveraging these observations. We also prove that the sampling error is theoretically bounded by the given error bound.

## III. OBSERVATIONS

In this paper, we define ML workloads on GPUs as workloads that utilize ML models, encompassing both inference and training processes. Given the widespread adoption of Python libraries in the ML field, we particularly focus on ML workloads that utilize widely-used libraries such as Tensor-Flow [11] and PyTorch [32], which leverage NVIDIA GPU libraries such as cuDNN, cuBLAS, etc., under the abstraction layer. While there are some pure C++/CUDA implementations of ML models [3], [30], as they are not as commonly used compared to other Python implementations, we focus on the Python-front-end and NVIDIA library-based back-end ML workloads.

### A. High Homogeneity in ML Workloads

Modern ML models often employ a repeated block structure across multiple layers. For example, ResNet50 [14] comprises 48 identical convolutional layers, while transformer-based LLMs consist of multiple repetitions of the transformer blocks that consist of a self-attention layer followed by fully connected layers. Moreover, ML workloads typically involve running the same model multiple times for batch iterations. Consequently, we can expect ML workloads to exhibit homogeneity at the GPU kernel level as well.

By employing a GPU hardware profiler such as Nsight Systems [31], we can unveil the kernel calls of ML workloads inside NVIDIA's CUDA libraries. Table II presents an example of a list of kernels alongside the number of calls made throughout the entire ResNet50 workload (refer to Table III for more details). We observe that in ML workloads, the kernels exhibit high repetition, as evidenced by the large number of kernel calls.

TABLE II

TOP 5 TIME-CONSUMING GPU KERNELS IN RESNET50 [14] WORKLOAD.
THE WORKLOAD INVOLVES A LARGE NUMBER OF KERNEL CALLS, AND
EACH KERNEL IS OPTIMIZED FOR "VOLTA" GPU ARCHITECTURE.

| Kernel Name | # Calls | Total Time (ns) |
|---|---|---|
| cudnn_infer_volta_scudnn_winograd_128x... | 19625 | 1185625785 |
| explicit_convolve_sgemm | 3925 | 964880834 |
| cudnn_infer_volta_scudnn_winograd_128x... | 7850 | 897755249 |
| volta_sgemm_128x64_nn | 23550 | 709594145 |
| winograd::generateWinogradTilesKernel | 7850 | 595149925 |

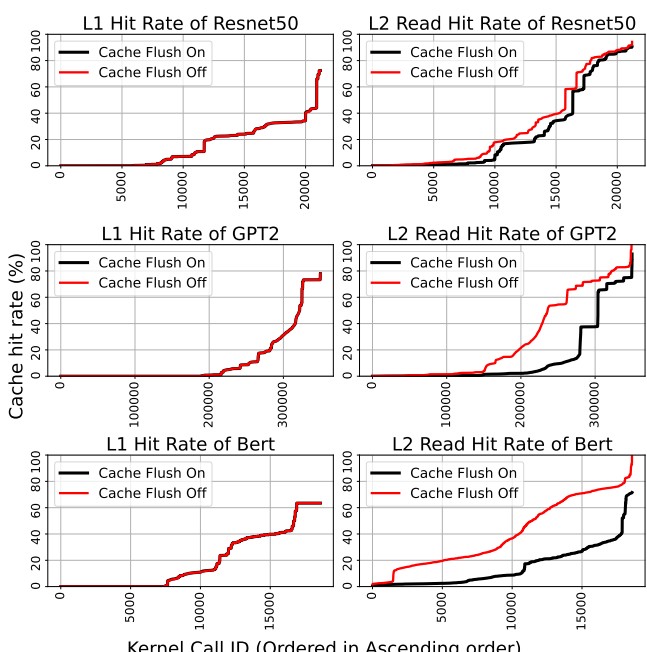

Fig. 1. L1 and L2 cache hit rate in Bert and ResNet50 workload, with cache flushing between kernel calls on/off.

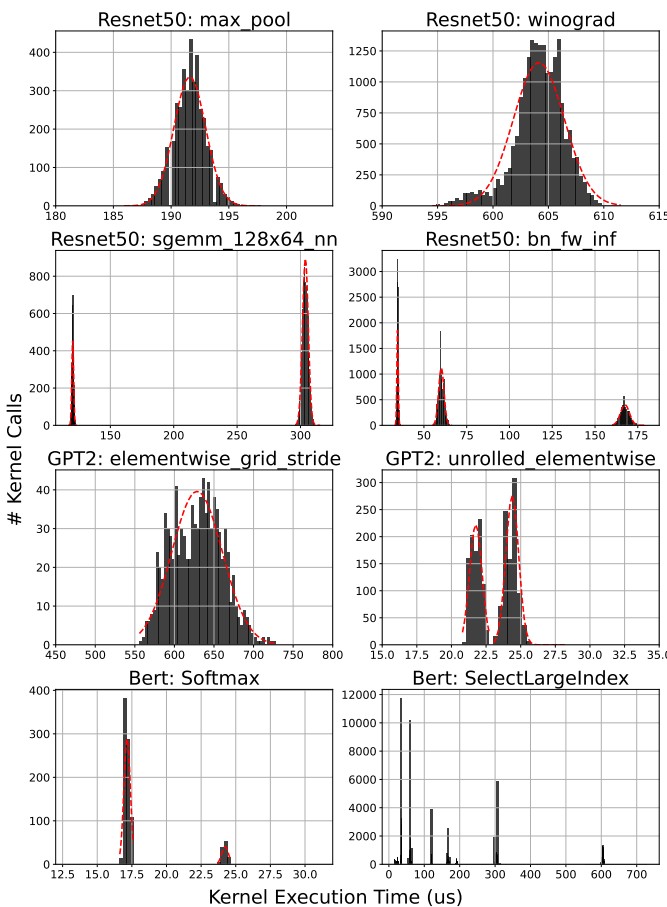

Fig. 2. Kernel execution time histograms of GPU kernels in ResNet50, GPT-2, and Bert workload. The workload and kernel name are on the top of each graph. The red dotted line shows the ideal normal distribution. We can observe very narrow execution time distributions in most of the GPU kernels.

Moreover, it is well known that the kernel names suggest they are highly optimized—some hand-coded, architecture-specific, and compiler-optimized—for particular GPU architectures, such as the Volta architecture GPU from NVIDIA, as indicated by the presence of "volta" in the kernel names [9]. This indicates that we should focus on the statistics that we can collect at runtime, unlike previous works [2], [19], [28] where they only collected statistics that are independent of architecture, such as the number of dynamically executed instructions or the number of memory requests.

### B. Cache-Unfriendly Nature of ML Workloads

We also observe that ML workloads do not benefit much from the GPU cache system between the kernel exit and the kernel call. This means that since the memory footprint of GPU kernels in ML workloads is very big, GPU kernels make no use of the cache when they are launched because for the following memory accesses the kernel will only experience cache misses.

To investigate this behavior, we conducted an experiment to compare the effect of the cache system in a GPU, with the ResNet50, GPT-2, and Bert workloads (refer to Table III). We measured the L1 and L2 cache hit rates under two conditions: 1) when the GPU cache is flushed between every kernel call, and 2) when the GPU cache remains intact (baseline).

Figure 1 depicts the L1 and L2 cache hit rates of each kernel called within the Bert and ResNet50 workloads. Each kernel's cache hit rate is sorted in ascending order for better visualization.

We observe that the L1 cache hit rate shows a very small difference between the two cache scenarios, where the cache flush between the kernel calls is On/Off. In the case of ResNet50, even the L2 hit rates show a similar trend, suggesting that the L2 cache does not significantly contribute to running the workload.

These results suggest that the majority of cache hits occur due to memory access locality within the kernel itself, rather than due to the locality between kernels. Consequently, the GPU cache does not substantially benefit consecutive kernels by leveraging data stored in the cache to reduce execution times.

Although some workloads show a difference in L2 cache statistics, we empirically discovered that such a difference in L2 cache hit rate does not greatly affect the execution time

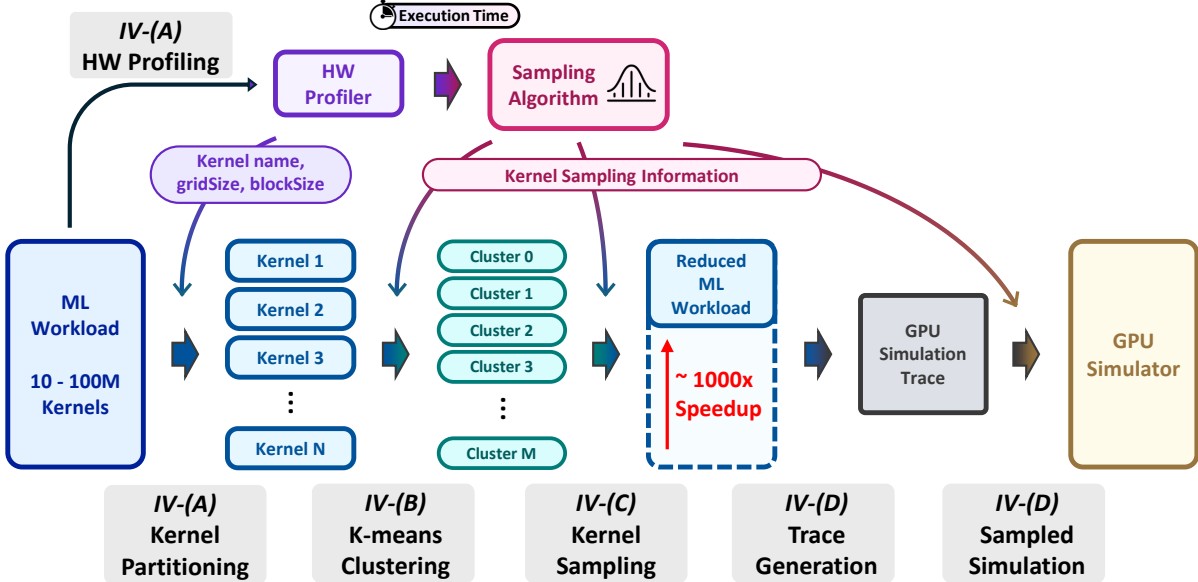

Fig. 3. Allegro's GPU kernel sampling methodology.

distribution of kernels, due to the long latency of L2 memory in GPUs. We will discuss about this issue more in the section III-C.

Hence, we can infer that the execution times of kernels are independent, as the kernels' execution order does not significantly affect subsequent kernel execution times. Additionally, even if the same kernel is launched multiple times, the cache's impact on kernel execution is minimal, resulting in uniform execution times irrespective of launch location or preceding kernels. In summary, this implies that kernels launched multiple times exhibit identically and independently distributed (i.i.d.) execution time.

### C. Analysis on Kernel Execution Time Histogram

We utilize Nsight-Systems to visualize the distribution of kernel execution times. We first partition the profiled kernels into kernel groups based on their name, blockSize, and grid-Size, so that we can plot a histogram of execution time. We observed that the histograms exhibit 1) a single narrow peak with a very small standard deviation divided by the mean ($\sigma/\mu$), or 2) multiple discrete peaks of narrow distributions. The example histograms are shown in Figure 2.

Since most of the GPU kernels' execution time distribution in the histogram has discrete and spread-out peaks, we can employ 1-D k-means clustering to separate them into subgroups, ensuring that each group contains only one narrow peak in the histogram. Then, since each kernel subgroup comprises a high number of kernels with very narrow distributions, we can apply the sampling methodology to significantly shorten simulation times while maintaining high accuracy.

In Section IV, we will demonstrate how we can exploit the fact that the execution times of kernels in a subgroup are identically and independently distributed (i.i.d.) by leveraging

the Central Limit Theorem (CLT). We will introduce a measure named $m_{min}$, representing the minimum number of samples required to ensure the error bound. Furthermore, we will outline an algorithm with a scheme for sampling the kernels to accurately predict the total execution time.

Although we observed the effects of the L2 cache in some workloads like GPT-2 and BERT (Figure 1), we could not observe the impact of L2 caches in the execution time histogram (Figure 2), as the histogram of GPT-2 and Bert were not very different from the ones of ResNet50. As shown in the histograms of all three workloads, kernel execution time distributions were very discrete and narrow. Therefore, we assumed that such L2 cache hit rate discrepancy in Figure 1 is negligible for our method of using execution time statistics in sampling.

Although we observed the effects of the L2 cache in some workloads like GPT-2 and BERT (Figure 1), these effects were not evident in the execution time histograms. The histograms for GPT-2 and BERT were not significantly different from those of ResNet50. As shown in the histograms for all three workloads, kernel execution time distributions were very discrete and narrow. Therefore, we assumed that the L2 cache hit rate discrepancy in Figure 2 is negligible for our method of using execution time statistics in sampling.

### IV. Allegro's Methodology

Figure 3 is an overall diagram of how Allegro performs kernel sampling to accelerate the GPU simulation. We first cluster the GPU kernels into groups based on their name, grid size, and block size. Then, we further subdivide each group with 1-D k-means clustering so that every kernel within a group exhibits a homogeneous execution time.

We sample $m_{min}$ kernels from each group, where the value of $m_{min}$ varies for each group. This $m_{min}$ value is a number that is obtained from the error bound calculation, such that the total estimated execution time falls within a specified error bound. The exact equation and the theory behind it are shown in Theorem 2. Following the sampling process, we generate simulation traces only for the sampled kernels and execute the simulation accordingly. In this way, we can drastically reduce the number of GPU kernels in a GPU simulation, achieving a high degree of speedup.

We adopt a statistical approach to propose the method for clustering, aiming to ensure that the sampled simulation result closely approximates the outcome of the full simulation within a small error bound.

### A. GPU Kernel Profiling and Partitioning

First, we profile the provided ML workload using a GPU hardware profiler, extracting various statistics including execution time, kernel name, block size, and grid size for each kernel. We then partition the kernels into groups based on their kernel name, block size, and grid size. Subsequently, we calculate the mean and variance of execution times for each kernel group.

### B. GPU Kernel Clustering

From our observation, we noticed that some kernel groups have multiple peaks in the execution time histogram, which means that even if the kernel has the same name and gridblock size, each kernel may have been used in a different context of the workload. Therefore, we need to split the kernel group into subgroups with homogeneous distribution.

To provide a metric for whether a group of kernels needs to be split or not, we exploit the Central Limit Theorem (CLT). The number of samples that we should take depends on the variance of execution time relative to the mean. The following section describes how we designed this metric, so we can recursively apply 1-D K-Means clustering with $k = 2$ to split the kernels afterwards.

Let $K$ be an arbitrary set of $N$ GPU kernels with the same name and grid/block size, and let $\mu$ and $\sigma^2$ be the mean and variance of the execution time of kernels in $K$. Assume we are sampling $m$ kernels from $K$. For any given $i \in \{1, ..., m\}$, we define a random variable $X_i$ to be the execution time of $i$-th sampled kernel.

We can assume that the sequence of random variables $\{X_1, X_2, ..., X_m\}$ are independent and identically distributed (i.i.d.) because we observed that the impact of GPU caches are negligible in ML workloads and thus the execution time of a GPU kernel is uniform throughout the full run. We predict the sum of the execution time of every kernel in $K$ as $N\bar{X}$ where $E[X_i] = \bar{X}$. To calculate the error between $\mu$ and $N\bar{X}$, we use the CLT.

**Theorem 1** (Central Limit Theorem). *Let $\{X_1, X_2, ..., X_m\}$ be a sequence of $m$ i.i.d. random variables having a distribution with mean $\mu$ and variance $\sigma^2$. Then, the sampled mean $\bar{X} = \frac{1}{m}\Sigma_{i=0}^{m} X_i$ converges to a random variable having*

---

**Algorithm 1** GPU Kernel Partitioning and Clustering

1: (input) $K_0$: List of Kernels from the workload
2: (input) $\epsilon$: Error bound for sampling, default = 5%.
3: (input) $m_{Th}$: Threshold value compared with $m_{min}$ to determine whether to cluster the group or not. default = 50.
4: (output) $C$: Group of kernel groups, the output of this algorithm.
5: **function** ALLEGRO()
6:     $M$: Map from keys to kernel groups. Global Instance.
7:     PARTITION_KERNEL()
8:     **for** (key, $K_i$) $\in M$ **do**
9:         CLUSTER_KERNEL($\{K_i\}$)
10:     **end for**
11: **end function**
12: **function** PARTITION_KERNEL()
13:     **for** $k \in K_0$ **do**
14:         key $\leftarrow$ strcat($k$.name,$k$.gridSize,$k$.blockSize)
15:         $M$[key].add($k$)
16:     **end for**
17: **end function**
18: **function** CLUSTER_KERNEL($K$) // $K$: group of kernel groups
19:     **for** $K_i \in K$ **do**
20:         $\mu \leftarrow$ mean($K_i$), $\sigma \leftarrow$ stdev($K_i$)
21:         $m_{min} \leftarrow max\{\lceil (1.96\sigma/\mu\epsilon)^2 \rceil, 30\}$
22:         **if** $m_{min} \leq m_{Th}$ **then** $C$.add($K_i$)
23:         **else** $\{K', K''\} \leftarrow$ kmeans_clustering($K_i$, k=2)
24:             CLUSTER_KERNEL($\{K', K''\}$)
25:         **end if**
26:     **end for**
27: **end function**

---

*a normal distribution with mean $\mu$ and variance $\sigma^2/m$ as $m \to \infty$.*

By using CLT and Lemma 4 in the Appendix (Sec. VIII-A), we can approximate that $N\bar{X}$ follows a normal distribution $N(N\mu, N^2\sigma^2/m) := N(\tilde{\mu}, \tilde{\sigma}^2)$. Then, we can use Theorem 2 below to obtain $m_{min}$, the minimum number of samples to ensure the error of $N\bar{X}$ is within the error bound $\epsilon$ under 95% confidence.

**Theorem 2.** *For a random variable $N\bar{X}$, which follows a normal distribution of $N(N\mu, N^2\sigma^2/m) := N(\tilde{\mu}, \tilde{\sigma}^2)$, the minimum number of samples to ensure the error between $N\bar{X}$ and $\tilde{\mu}$ is smaller than the error bound $\epsilon$ for 95% confidence is as follows:*

$$m_{min} := max\left\{ \left\lceil \left(\frac{1.96}{\epsilon}\frac{\sigma}{\mu}\right)^2 \right\rceil, 30 \right\}.$$

*Proof.* Proof in the Appendix (Section VIII-A). $\square$

We utilize $m_{min}$ as a metric for each kernel group to assess whether the kernels within the group exhibit homogeneous execution times. The overall algorithm for both the partitioning and clustering process is depicted in Algorithm 1.

### C. GPU kernel sampling

To predict the total execution time of the whole workload, we apply Algorithm 1 to every kernel group in the workload. Assume we split kernels into $k$ groups $\{K_1, K_2, ..., K_k\}$, and we sample $m_1, m_2, ..., m_k$ kernels from each group, using the

error bound $\epsilon$ for all groups. Each $m_i$ is derived from the $m_{min}$ of Theorem 2.

Let $Y^*$ be the ground-truth total execution time, $\bar{X}_i$ be the average execution time of the sampled kernels from $K_i$, and $N_i$ be the number of kernels in $K_i$. Note that $\bar{X}_i$'s are normal and mutually independent, i.e., $\bar{X}_i \sim N(\mu_i, \sigma_i^2)$ for $\forall i \in \{1, ..., k\}$. Then, we predict the total execution time as $Y = \sum_k^{i=1} N_i \bar{X}_i$, and according to Lemma 4, the random variable $Y$ also follows a normal distribution, i.e., $Y \sim N(\sum_{i=1}^k N_i \mu_i, \sum_{i=1}^k N_i^2 \frac{\sigma_i^2}{m_i})$.

In Theorem 3, we ensure that the error of our prediction also falls within the same error bound $\epsilon$ with 95% certainty.

**Theorem 3.** *If the error between $N_i \bar{X}_i$ and $N_i \mu_i$ is bounded by $\epsilon$ with 95% confidence, the error between $Y$, the prediction for total execution time, and its ground-truth $Y^* = \sum_{i=1}^k N_i \mu_i$, is also bounded by $\epsilon$.*

*In other words, the error $e$ between $Y$ and $Y^*$ is constrained by the following inequality:*

$$e = \left| \frac{Y - Y^*}{Y^*} \right| = \left| \frac{1.96\sqrt{\sum_{i=1}^k N_i^2 \frac{\sigma_i^2}{m_i}}}{\sum_{i=1}^k N_i \mu_i} \right| \le \epsilon.$$

*Proof.* Proof in the Appendix (Section VIII-A). □

### D. Trace Generation and Simulation Execution

We implemented Algorithm 1 in conjunction with Macsim's trace generation tool [7] to produce traces for the sampled GPU simulation. Macsim is capable of executing NVIDIA GPU simulations with SASS-Assembly in trace-driven mode. By utilizing the sampled trace alongside the sampling information, we generated simulation statistics to predict the outcome of the original workload. All seven latest ML workloads listed in Table III were successfully executed on Macsim. Detailed performance results and error analysis will be discussed in the evaluation section.

## V. EVALUATION

### A. Experiment Environment Setups

We used NVIDIA RTX 2080 GPU to evaluate our sampling methodology along with Nsight-systems [31] as GPU hardware profiler. Table III shows the list of ML workloads that are used in this paper. For more information about the workloads, check the Appendix (Section VIII-B).

### B. Allegro's Pre-processing Overhead

The performance overhead introduced by Allegro is negligible and operates as a one-time cost. While hardware profiling and sampling indeed add performance overhead, they are only incurred during trace generation for GPU simulation and are not incurred during actual simulation execution.

In comparison to the time required for trace generation, the time taken for GPU hardware profiling is negligible, being over x1000 less than the trace generation. Additionally, the sampling process itself is also very efficient—we conducted tests on workloads consisting of several million to 50 million

TABLE III
LIST OF ML WORKLOADS USED IN THIS PAPER, ALONG WITH THE NUMBER OF GPU KERNELS AND A SHORT DESCRIPTION.

| Name | # Kernels | Workload Description |
|---|---|---|
| Bert | 1858800 | Performing sequence classification on 10,000 premise/hypothesis pairs using the BERT-Medium-MNLI model. |
| Bloom | 51834362 | Generating 1,000 sentences, each with a length of 100 tokens, using the Bloom model. |
| Deit | 792850 | Classifying 3,925 ImageNet datasets using the Data-efficient image Transformer (DeiT) model. |
| Gemma | 9079126 | Generating 1,000 sentences, each with a length of 100 tokens, from the GEMMA language model. |
| GPT-2 | 34981000 | Generating 1,000 sentences, each with a length of 100 tokens, from the GPT-2 model. |
| Olmo-bitnet | 2544766 | Generating 10 sentences, each with a length of 100 tokens, from the OLMo-Bitnet language model. |
| ResNet50 | 2812741 | Classifying 13,400 ImageNet datasets using the ResNet50 model. |

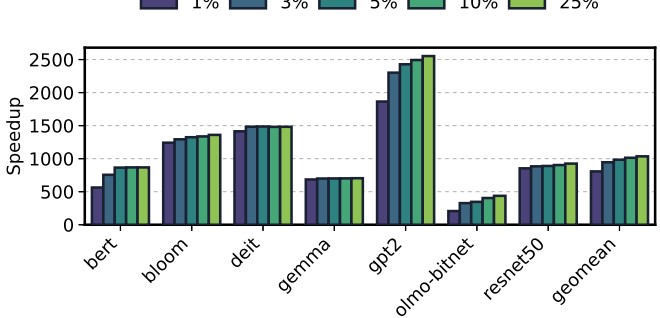

Fig. 4. Allegro's speedup for GPU simulations, using different error bounds from $\epsilon = 1\%$ to $\epsilon = 25\%$.

kernel calls, and the splitting, clustering, and sampling procedure only took a few minutes to complete. As the sampling information file is generated alongside the simulation traces, there is no additional performance overhead during GPU simulation execution.

### C. Speedup and Error Validation of Allegro

First, we evaluated how the error bound $\epsilon$ affects the speedup of the GPU simulation. We varied the value of $\epsilon$ to 1%, 3%, 5%, 10%, and 25% and measured the speedup for each ML workload.

Figure 4 illustrates how the speedup value changes as the error bound is increased. We observe that when $\epsilon = 5\%$, Allegro can achieve a geometric mean speedup of 983.96x in the given workload simulations, or even more when a larger error bound is used. It is also important to note that the speedup does not scale linearly due to the requirement of sampling at least 30 kernels to satisfy the CLT (Theorem 1).

Figure 5 shows the measured error for each given error bound $\epsilon$. When $\epsilon = 5\%$, Allegro achieves a geometric mean error of 0.057% in predicting the total execution time of the workloads.

Note that the actual error of the total execution time is considerably smaller than the specified error bound $\epsilon$. Also,

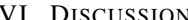

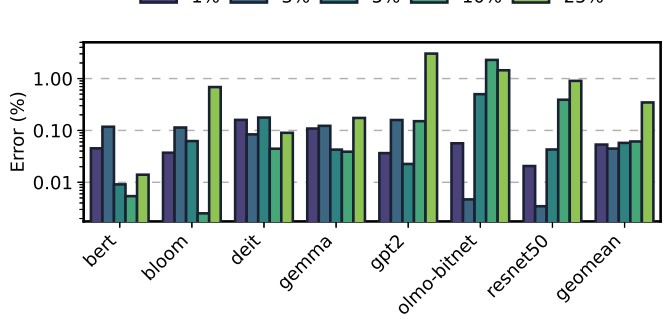

Fig. 5. Measured error of Allegro for various error bounds from $\epsilon = 1\%$ to $\epsilon = 25\%$.

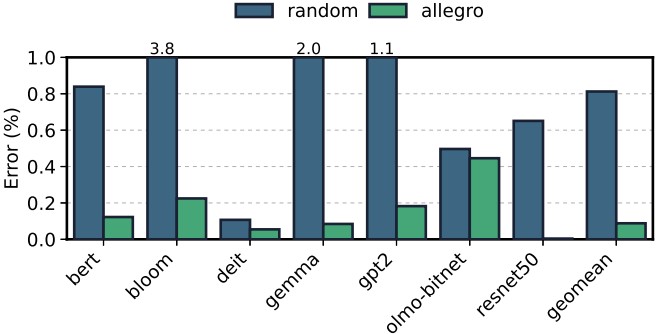

Fig. 6. Error validation for various GPU Simulation optimization methods.

since the actual error is much smaller than the error bound, increasing the error bound does not significantly affect the actual error. This is because the randomness involved in the clustering stage accounts for most of the actual error. This suggests that there still exists more room to reduce the number of samples, thereby providing more opportunity to enhance the amount of speedup achieved.

We also compared the accuracy of Allegro against other sampling methods. For the baseline, we employed random kernel sampling, where we randomly sampled kernels from the entire workload until we achieved the same degree of speedup as Allegro. For instance, if Allegro achieved a 100x speedup on a certain workload, the baseline will randomly sample kernels until the sum of the execution times of the sampled kernels is 1% of the total execution time of the whole workload. Since the baseline method does not utilize information from the profiler as Allegro does, we expect the random sampling to have a larger error than Allegro.

We compare the error of the two sampling methods while constraining the amount of speedup. For Allegro, we set $\epsilon = 5\%$, and the random sampling method also aimed for the same speedup achieved by Allegro with this error bound.

We can observe from Figure 6 that Allegro achieves the highest accuracy on a constrained amount of speedup. In this experiment, Allegro achieved a geomean error of 0.088%, whereas the random sampling achieved a geomean error of 0.81%, which is 9.22 times bigger than our method.

## VI. DISCUSSION

### A. Limitations of GPU Kernel Sampling

We examined scenarios in which the sampling method of Allegro may not achieve notable speedup or where its error could surpass the theoretical prediction. In this section, we provide three categories of workload scenarios where Allegro may encounter its limitations.

**Cases of Insufficient Speedup:** In certain scenarios, Allegro, or the GPU kernel sampling methodology in general, may not achieve a satisfactory level of simulation speedup. If the number of samples is relatively too large compared to the number of kernels in the entire workload, or if the workload itself contains a very small number of kernels, Allegro may not be able to achieve the high degree of speedup observed in ML workloads.

For example, in non-ML GPU workloads like the Rodinia benchmark suite [6], benchmarks such as `backprop` or `lavaMD` comprise fewer than 10 GPU kernels. In such cases, Allegro cannot generate a significant level of speedup in GPU simulations. This is because as shown in Theorem 2, Allegro requires more than 30 samples of kernels from each kernel cluster. Thus, kernel clusters containing fewer than 30 kernels would not benefit from Allegro's sampling approach.

However, as Allegro operates orthogonal of other simulation-speedup techniques, such as early stopping in PKA [2] or the use of analytical models in GPUMech [18], there remains further potential to accelerate GPU simulation even after the adoption of Allegro in the simulators.

**Cases of Non-i.i.d. Kernels:** Allegro's sampling methodology runs under the assumption that the execution time of kernels is independent and identically distributed (i.i.d.). In certain GPU workloads, this assumption may not hold, as the performance of GPU kernels may be influenced by other factors such as GPU cache.

We've identified two scenarios where we cannot guarantee the independent and identically distributed (i.i.d.) nature of kernel execution times. First, the independence assumption of kernel execution times may be compromised when the execution time of each kernel is influenced by preceding kernels due to GPU caches, leading to inter-dependencies in execution times between kernels. Second, contention on inter-connections between other components such as DRAM, CPU, or other GPUs in multi-GPU systems may result in varying execution times depending on the workload phase. Although these scenarios didn't manifest in our tested ML workloads due to their homogeneity and single-GPU environment, it's essential to further investigate how we can quantify workload homogeneity to ensure the i.i.d. nature of kernels.

**Reduced number of memory requests:** In scenarios where GPU simulations require full memory traces, kernel sampling methods may not be a suitable way to expect precise simulation results. For instance, when employing ML workloads to analyze how the workload populates architectural components such as memory, cache, TLB, etc., sampling the small number of kernels and only running them may not let the simulator

adequately capture the memory access patterns. This limitation arises due to the significantly reduced number of memory requests compared to running the full workload, potentially leading to incomplete or inaccurate insights into the architectural behavior. Therefore, excessively speeding up simulations using Allegro may compromise the fidelity of the detailed behaviors expected in the simulation.

### B. Future works

**Leveraging Analytical Models for Further Speedup:** Designing an analytical model to reduce the workload size after sampling may be a good follow-up work after Allegro. We can further exploit homogeneity and the statistics obtained from hardware profiling to design such an analytical model. Since there exist similar previous works [2], [19] that utilize methods to accelerate simulation after applying sampling, Allegro can also benefit from analogous approaches.

**Power estimation on RTL simulations:** Workload sampling can offer significant benefits for tasks such as power estimation, particularly in the context of RTL design and simulations. For instance, collecting statistics related to power usage often doesn't necessitate a full workload simulation. By deploying a methodology similar to Allegro on RTL simulators, power measurements can be conducted more efficiently on RTL design and optimization processes.

## VII. CONCLUSION

We present Allegro as a novel GPU kernel sampling approach for accelerating GPU simulations on ML workloads characterized by homogeneity and repetition. By utilizing statistical methods to design a sampling scheme, Allegro achieved an average speedup of 983.96x on 7 of the latest ML workloads with an error rate of 0.057%, and achieved 9.22x smaller error in the comparison with the random sampling method. Despite Allegro's limitations with non-uniform or non-ML workloads, its robust mathematical foundation, compatibility with other acceleration methods, and substantial performance improvement set the stage for Allegro for broader adoption and application.

## VIII. APPENDIX

### A. Proofs for Theorems and Lemmas

**Theorem 2.** *For a random variable $N\bar{X}$ which follows a normal distribution of $N(N\mu, N^2\sigma^2/m) := N(\tilde{\mu}, \tilde{\sigma}^2)$, the minimum number of samples to ensure the error between $N\bar{X}$ and $\tilde{\mu}$ is smaller than the error bound $\epsilon$ for 95% confidence is as follows:*

$$m_{min} := max\left\{\left\lceil\left(\frac{1.96}{\epsilon}\frac{\sigma}{\mu}\right)^2\right\rceil, 30\right\}.$$

*Proof.* If we assume normal distribution for $N\bar{X}$, error bound of $N\bar{X}$ for 95% confidence is calculated as $P(\tilde{\mu} - 1.96\tilde{\sigma} \leq N\bar{X} \leq \tilde{\mu} + 1.96\tilde{\sigma}) = 0.95$.

If we aim for a 5% bound on the error $e := \left|\frac{(N\bar{X}-\tilde{\mu})}{\tilde{\mu}}\right|$ with the same level of certainty, we can derive the following inequality:

$$e = \left|\frac{(N\bar{X}-\tilde{\mu})}{\tilde{\mu}}\right| = \left|\frac{(\tilde{\mu}+1.96\tilde{\sigma})-\tilde{\mu}}{\tilde{\mu}}\right| = 1.96\frac{\sigma}{\mu\sqrt{m}} \leq 0.05,$$

and therefore $m$ should satisfy the following condition:

$$m \geq \left(\frac{1.96}{0.05}\frac{\sigma}{\mu}\right)^2. \tag{1}$$

Inequality (1) suggests that if the number of samples $m$ is large enough such that the inequality holds, the error between $\tilde{\mu}$ and $N\bar{X}$ is less than 5% for 95% chance.

It is empirically well-known that the Central Limit Theorem (CLT) becomes useful when the sample size $m$ is equal to or greater than 30 [38]. Hence, we define $m_{min}$, the minimum number of samples required to ensure the error bound, as Equation (2) with an arbitrary error bound $\epsilon > 0$.

$$m_{min} := max\left\{\left\lceil\left(\frac{1.96}{\epsilon}\frac{\sigma}{\mu}\right)^2\right\rceil, 30\right\} \tag{2}$$

We also apply a ceiling function to ensure $m_{min}$ is a natural number. □

**Theorem 3.** *If the error between $N_i\bar{X}_i$ and $N_i\mu_i$ is bounded by $\epsilon$ with 95% confidence, the error between $Y$, the prediction for total execution time, and its ground-truth $Y^* = \sum_{i=1}^{k} N_i\mu_i$, is also bounded by $\epsilon$.*

*In other words, the error $e$ between $Y$ and $Y^*$ is constrained by the following inequality:*

$$e = \left|\frac{Y-Y^*}{Y^*}\right| = \left|\frac{1.96\sqrt{\sum_{i=1}^{k} N_i^2\frac{\sigma_i^2}{m_i}}}{\sum_{i=1}^{k} N_i\mu_i}\right| \leq \epsilon.$$

*Proof.* For $\forall i \in \{1, ..., k\}$, $m_i \geq (\frac{1.96}{\epsilon}\frac{\sigma_i}{\mu_i})^2$ from Inequality 1.

Then, by reordering each side we can get $\mu_i^2 \geq (\frac{1.96}{\epsilon})^2\frac{\sigma_i^2}{m_i}$.

By summing up the both side for all $i$'s from 1 to $k$, we obtain $\sum_i N_i^2\mu_i^2 \geq (\frac{1.96}{\epsilon})^2 \sum_i \frac{N_i^2\sigma_i^2}{m_i}$.

Since $N_i \geq 0$ and $\mu_i \geq 0$, $(\sum_i N_i\mu_i)^2 \geq \sum_i N_i^2\mu_i^2$ holds.

Thus, $(\sum_i N_i\mu_i)^2 \geq (\frac{1.96}{\epsilon})^2 \sum_i \frac{N_i^2\sigma_i^2}{n_i}$.

Taking the square root on each side preserves the inequality direction because both sides are non-negative.

$$\sum_i N_i\mu_i \geq \frac{1.96}{\epsilon}\sqrt{\sum_i \frac{N_i^2\sigma_i^2}{n_i}}$$

Given that $\epsilon > 0$, we can obtain the original inequality, which infers that the error $e$ is bounded by the error bound $\epsilon$.

$$e = \left|\frac{1.96\sqrt{\sum_{i=1}^{k} N_i^2\frac{\sigma_i^2}{m_i}}}{\sum_{i=1}^{k} N_i\mu_i}\right| \leq \epsilon.$$

□

**Lemma 4.** *Let $\{X_1, X_2, ..., X_m\}$ be mutually independent normal random variables, i.e., $X_1 \sim N(\mu_1, \sigma_1^2)$, $X_2 \sim N(\mu_2, \sigma_2^2)$, and so on. Then, for any $a_1, a_2, ..., a_m \in \mathbb{R}$, the linear combination $Y = a_1 X_1 + a_2 X_2 + ... + a_m X_m$ also follows a normal distribution of*

$$Y \sim N(\Sigma_{i=1}^m a_i \mu_i, \ \Sigma_{i=1}^m a_i^2 \sigma_i^2).$$

*Proof.* A set of $n$ independent normal random variables is equivalent to an $n \times 1$ vector having a multivariate normal distribution with a diagonal covariance matrix. We define a random vector $\mathbf{x}$ as follows: $\mathbf{x} = [X_1 \ X_2 \ ... \ X_n]^T \sim N(\mu, \Sigma)$ where $\mu = [\mu_1 \ \mu_2 \ ... \ \mu_n]^T$ and $\Sigma = diag(\sigma_1^2, \sigma_2^2, ..., \sigma_n^2)$.

Let a constant matrix $A = [a_1 \ a_2 \ ... \ a_n]$ and a vector $\mathbf{b} = \mathbf{0}$. Then, by applying the linear transformation theorem for the multivariate normal distribution [1], [36], the linear combination $Y = A\mathbf{x} + \mathbf{b}$ also follows a normal distribution, i.e., $Y = A\mathbf{x} + \mathbf{b} \sim N(A\mu, A\Sigma A^T)$.

Therefore, $Y = A\mathbf{x} + \mathbf{b} = [a_1 \ .. \ a_n][X_1 \ ... \ X_n]^T + \mathbf{0} = \Sigma_{i=1}^n a_i X_i$ and

$$Y \sim N([a_1 \ ... \ a_n][\mu_1 \ ... \ \mu_n]^T,$$

$$[a_1 \ ... \ a_n]diag(\sigma_1^2, ..., \sigma_n^2)[a_1 \ ... \ a_n]^T)$$

$$= N(\Sigma_{i=1}^n a_i \mu_i, \Sigma_{i=1}^n a_i^2 \sigma_i^2).$$

$\square$

### B. ML Model Specifications

We downloaded every ML model in Table III from Huggingface [20]. Below are the URLs to Huggingface repositories of the models and datasets that we used for the ML workloads.

- Image Dataset [17]: We used Imagenette, a smaller subset of 10 easily classified classes from Imagenet [10]. Link: https://huggingface.co/datasets/frgfm/imagenette
- Bert [4]: Pytorch pre-trained model converted from the official Google BERT repository's tensorflow implementation. Trained on MNLI. Link: https://huggingface.co/prajjwal1/bert-medium-mnli
- Bloom [35]: 8-bit quantized BLOOM model. Link: https://huggingface.co/ybelkada/bloom-1b7-8bit
- Deit [40], [41]: Small sized Data-efficient Image Transformer. Link: https://huggingface.co/facebook/deit-small-distilled-patch16-224
- Gemma [39]: 2B base version of the Gemma model by Google. Link: https://huggingface.co/google/gemma-2b
- GPT-2 [34]: GPT-2 model by OpenAI. Link: https://huggingface.co/openai-community/gpt2
- Olmo-bitnet [12]: 1-bit transformer [26] implementation of Olmo (Open Language Model). Link: https://huggingface.co/NousResearch/OLMo-Bitnet-1B
- ResNet50 [14]: Convolutional neural network with residual blocks. Link: https://huggingface.co/microsoft/resnet-50

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
