# OpenReview forum: "Allegro: GPU Simulation Acceleration for Machine Learning Workloads"
_iscaconf.org/ISCA/2024/Workshop/MLArchSys — MLArchSys 2024 OralPoster_

### Official Review · Reviewer_T3rp · 2024-05-23
**Simple approach with theoretical guarantee and great results. Evaluation can be improved.**

**Confidence:** 4
**Rating:** 6

**Detailed Feedback And Questions For Authors:**

The paper shows excellent results compared to existing works in terms of both accuracy and speedup. The simulation errors are extremely small: less than 1% in most cases. The simulation time speedup over existing systems is impressive. The paper also builds on a nice insight that most GPU kernels in ML workloads are independent of other kernels. Therefore, we can measure the runtime of each kernel independently (no need to set up a context). The proposed approach is simple, sampling based on the Central Limit Theorem, which also provides a theoretical guarantee on an error bound.

One concern with this approach is that although it doesn’t require hardware during the simulation execution, it still requires hardware to sample the kernels’ runtime in advance. If most workloads share the same common sets of GPU kernels, then we can pre-sample all common kernels. However, if a workload includes kernels that are not pre-sampled, then this approach wouldn’t work. While in the past, most frameworks (PyTorch and TensorFlow) depends on CuDNN/CuBLAS kernels, they now move to a code generation strategy (e.g. PyTorch 2.0 generates Triton kernels). In this paradigm, there is a high chance that generated kernels (arbitrary fusions of multiple ops) won’t be part of the pre-sampled set.


The major weakness of the paper is that only workloads on a single device are evaluated. This means that the majority of the kernels are compute kernels, and there is no communication operation. We know that communication and compute kernels utilize dramatically different sets of resources and exhibit very different characteristics. Communication is also required in the most capable models like LLMs. Therefore, for the simulation to be useful, it also has good accuracy on communication kernels.

While the paper is mostly well written, it misses some important technical details. The simulation execution is extremely short, and it doesn’t explain at all how the simulation execution utilizes the pre-sampled data. I assume that all kernels must be profiled offline in advance, but the paper doesn’t explicitly mention that. Next, during the simulation, does the simulator then sample a runtime of the kernel from the modeled Gaussian distribution. In the case where a single kernel has multiple clusters (e.g. volta_sgemm_128x64_nn), how does the simulator decide which cluster to use during the simulation?

The random kernel sampling experiment requires more explanation. I don’t understand the setup and how it is different from the proposed method. I also don’t understand the discussion around the case of insufficient speedup, perhaps because I don’t understand how the simulation execution actually utilizes the samples. If sampling is done offline (before simulation), then there should be any slowdown using this method.

**Top Reasons To Accept The Paper:**

* Great results in terms of accuracy and speedup
* Provide nice insights (iid behavior)
* Simple approach with a theoretical guarantee on error bound

**Top Reasons To Reject The Paper:**

* Applicability of the approach to workloads with generated fused kernels
* Evaluation on only single-device workloads (no evaluation on communication kernels)
* Missing some technical details

---

### Official Review · Reviewer_Rykn · 2024-05-24
**Interesting paper on static sampling method to optimize GPU simulation**

**Confidence:** 3
**Rating:** 6

**Detailed Feedback And Questions For Authors:**

This paper presents very useful sampling optimization for GP-GPU simulation. This way, researchers and developers can cut/sample the number of to-be-simulated traces before simulation actually starts, and can gain a huge speedup for certain set of applications. Particularly, this method is based on observations that many ML workloads are pretty homogenous and cache-unfriendly, which includes well-known apps like Resnet and Gemma. Despite the fact, this solution might not be useful for more complex ML apps that employ elaborate optimization tricks, or for fetching more fine grain architectural performance counters, high-level observations can be very handy for early stage research and proof of hypothesis which are also very important.

**Top Reasons To Accept The Paper:**

1) Provides very useful sampling optimization to speedup GP-GPU simulation with relatively homogenous and cache-unfriendly, yet well-known ML apps like Gemma, Resnet and others in between.
2) Motivations, proves and explanations are clear.
3) Speedup numbers are impressive, even though use cases are limited.

**Top Reasons To Reject The Paper:**

1) As was mentioned by the paper itself, sampling is limited for homogenous and cache-unfriendly workloads only, hence project's impact is limited and might not be useful for more complicated, practical and exciting projects.
2) Accordingly, its potential for more discussions might be rather limited as per workshop goals.
3) Concerned about the idea of directly sourcing the latency numbers from other papers as a direct comparison, given setup, application and environment variables might be totally different.

---

### Official Review · Reviewer_1dm9 · 2024-05-26
**Good theoretical proofs for error bounds, but evaluation does not fully validate its usefulness/accuracy**

**Confidence:** 3
**Rating:** 5

**Detailed Feedback And Questions For Authors:**

1. How is faster simulation for LLMs helping? Isn't analytical modeling more convenient? Motivate your problem better
2. The cited analytical models are quite old and for generic applications. What about the newer analytical models such as "Performance Modeling and Scalability Optimization of Distributed Deep Learning Systems" or "AMPeD: An Analytical Model for Performance in Distributed Training of Transformers" that are specific for CNNs and transformers, respectively?
3. Is Allegro open-source?
4. I am not very sure when you say that the name of GPU architecture in the kernel names suggests that they are optimized for that GPU.  Is it a well-established fact from some source or just a guess?
5. In section III.B, the L2 hit rate for BERT is different for flush on and off. So that means Allegro is not valid for Bert, is that so?
6. Could you show the histogram for Bert kernels too in Figure 2 for more confidence?
7. Explain the pattern for Figure 5. Is there any pattern or correlation? When the error bound is lower, the measured error is higher for some workloads while opposite for other workloads. The trends are not the same for the workloads while it is not the case in Figure 4. Does this mean that the theoretical bounds are not accurate enough?
8. Figure 6 shows the error only in execution times of the workloads. Shouldn't the error be validated for other metrics too such as cache misses, hits, IPCs, memory accesses? The purpose of going for detailed simulation is to be able to predict these metrics otherwise one can go for analytical models, which are quite accurate in predicting the execution times.
9. On page 7, you say "Although these scenarios didn't manifest our tested ML workloads"  while it clearly appeared in BERT in figure 1. There was a dependence on GPU caches and hence IID can't be claimed. Is that right?

**Top Reasons To Accept The Paper:**

1. Very good theoretical proof for error bounds
2. Upto 1000x speedup in simulation time with minimal error.

**Top Reasons To Reject The Paper:**

1. Problem is not well motivated.  Why should I go for simulation of LLMs on GPUs? Isn't analytical model enough? Even in the results the error is shown for the execution time prediction and not for other useful metrics.

2. Not sure of the pattern in Fugure 5, hence I doubt the accuracy/usefulness of the theoretical proof

3. Evaluation needs to be stronger

---

### Official Review · Reviewer_XK3X · 2024-05-28
**Accelerating GPU simulation for ML workloads with statistical analysis**

**Confidence:** 4
**Rating:** 6

**Detailed Feedback And Questions For Authors:**

The paper clearly motivates the issue of slow GPU simulation for ML workloads and highlights the need for efficient acceleration techniques (with high accuracy). The proposed method, Allegro, uses statistical method, based on the homogeneity and i.i.d. nature of ML kernels which is a good contribution for the workshop. The method is supported by a strong theoretical foundation, including error bounds and analysis of kernel execution time distribution. Finally the results show promising results, high speedup with low error. Here are some questions/feedback:

(1) Could you elaborate on the limitations of Allegro in scenarios where the assumption of i.i.d. kernel execution times is not met? How does that impact the accuracy of your method? Is it possible to extend Allegro to handle these cases, perhaps by incorporating techniques for detecting and addressing non-i.i.d. behavior?

(2) I did not see comparison against other methods (speedup/error) in simulating ML workloads on GPUs.

(3) Do we need to do the entire sampling procedure for each GPU hardware + Compiler? How do you plan to handle differences in the GPU compilation workflow?

(4) It would be great if the authors consider releasing the code and scripts of Allegro.

**Top Reasons To Accept The Paper:**

++ The paper identifies an important problem in GPU simulation of ML workloads, that is slow simulation speed compared to real-world execution.

++ The proposed method, Allegro, leverages two insights: (1) the homogeneity and (2) i.i.d. nature of ML kernels and develop an efficient sampling mechanism. This sampling approach results in significant speedup (~1000$\times$).

++ The paper also provides theoretical foundation for the proposed sampling method, including proof of error bounds and analysis of kernel execution time distribution.

**Top Reasons To Reject The Paper:**

-- The paper attempts to bring up some of the limitations of Allegro, however, the paper does not discuss scenarios where kernels that are not truly i.i.d. or where the full memory traces are needed.

-- The experimental results show significant speedup, but the evaluation seems still limited. For example, it was not clear how training/inference would impact the results. Also, it could help to show comparison with some of the previous techniques that the authors mentioned in the paper.

-- (nit) The paper exceeds the page limit for the workshop.

---

### Decision · Program_Chairs · 2024-05-30

**Decision:**

Accept (Oral/Poster)

**Comment:**

Congratulations! We are pleased to inform you that your paper has been accepted for presentation at MLArchSys 2024. We look forward to your participation at the workshop. Further details regarding the schedule and format will be provided soon. See you at the workshop!